# Changes in clinical markers observed from pharmacist-managed cardiovascular risk reduction clinics in federally qualified health centers: A retrospective cohort study

Jasmine D. Gonzalvo[1][◉]*, Ashley H. Meredith[1][◉], Sonak D. Pastakia[1][◉], Michael Peters[2][‡], Madilyn Eberle[1][‡], Andrew N. Schmelz[3][‡], Lauren Pence[2][‡], Jessica S. Triboletti[3][‡], Todd A. Walroth[2][‡]

1 Purdue University College of Pharmacy, Indianapolis, Indiana, United States of America, 2 Eskenazi Health, Indianapolis, Indiana, United States of America, 3 Butler University College of Pharmacy and Health Sciences, Indianapolis, Indiana, United States of America

◉ These authors contributed equally to this work.
‡ MP, ME, ANS, LP, JST and TAW also contributed equally to this work.
* jgonzalv@purdue.edu

**Data Availability Statement:** All relevant data are within the paper and Supporting Information files.

## Abstract

### Background

Reductions in hemoglobin A1c (HbA1C) have been associated with improved cardiovascular outcomes and savings in medical expenditures. One public health approach has involved pharmacists within primary care settings. The objective was to assess change in HbA1C from baseline after 3–5 months of follow up in pharmacist-managed cardiovascular risk reduction (CVRR) clinics.

### Methods

This retrospective cohort chart review occurred in eight pharmacist-managed CVRR federally qualified health clinics (FQHC) in Indiana, United States. Data were collected from patients seen by a CVRR pharmacist within the timeframe of January 1, 2015 through February 28, 2020. Data collected include: demographic characteristics and clinical markers between baseline and follow-up. HbA1C from baseline after 3 to 5 months was assessed with pared t-tests analysis. Other clinical variables were assessed and additional analysis were performed at 6–8 months. Additional results are reported between 9 months and 36 months of follow up.

### Results

The primary outcome evaluation included 445 patients. Over 36 months of evaluation, 3,803 encounters were described. Compared to baseline, HbA1C was reduced by 1.6% (95%CI -1.8, -1.4, *p*<0.01) after 3–5 months of CVRR care. Reductions in HbA1C persisted at 6–8 months with a reduction of 1.8% ([95%CI -2.0, -1.5] *p*<0.01). The follow-up losses were 29.5% at 3–5 months and 93.2% at 33–36 months.

**Funding:** The author(s) received no specific funding for this work.

**Competing interests:** The authors have declared that no competing interests exist.

## Conclusions

Our study augments the existing literature by demonstrating the health improvement of pharmacist-managed CVRR clinics. The great proportion of loss to follow-up is a limitation of this study to be considered. Additional studies exploring the expansion of similar models may amplify the public health impact of pharmacist-managed CVRR services in primary care sites.

## Introduction

Globally, the number one cause of death is cardiovascular disease, which includes coronary artery disease, cerebrovascular disease, rheumatic heart disease, and additional conditions of the heart and blood vessels [1, 2]. In the United States alone, one in four deaths are attributed to heart disease [2]. From 2014 to 2015, healthcare expenditures related to heart disease cost the United States approximately $219 billion [2]. Cardiovascular diseases create a large financial, emotional, and physical burden for patients, health care workers, and systems. Risk factors, including medical conditions and lifestyle choices, have been identified to prevent the risk of developing cardiovascular disease, as studies have established a strong association between elevated blood glucose and cardiovascular complications [3]. By reducing the hemoglobin A1C (HbA1C) by one percentage point, epidemiologic research demonstrated an 18% reduction in combined fatal and nonfatal myocardial infarction [3]. Furthermore, each percentage point reduction in HbA1C (e.g., from 10% to 9%) results in an estimated savings associated with medical expenditures of $685-$950 per patient, per year [4].

One approach to addressing the high burden of chronic diseases to public health, including diabetes, has been to involve pharmacists in the care of patients within primary care settings. Pharmacists have become an integral component of cardiovascular disease state management in many Federally Qualified Health Centers (FQHC) [5–7]. FQHCs are primary care clinics that receive federal funding to provide healthcare to underserved communities [8]. Similar health centers provide care to almost 10% of Americans [9]. Common responsibilities of the pharmacist within the FQHC may include medication therapy management, transitions of care, cost containment, and medication access. Pharmacist-managed cardiovascular risk reduction (CVRR) clinics are a key factor in reducing health care costs and cardiovascular disease burden. These clinics address chronic disease states such as hypertension, diabetes, dyslipidemia, and smoking cessation [10–12]. Some evidence has established the impact of pharmacists in improving several cardiovascular clinical markers, such as reductions in HbA1C levels, systolic and diastolic blood pressure, and lipid levels [10–12]. These studies typically encompass six months to two years and do not focus on populations who are underserved. The result is a lack of literature surrounding the long-term cardiovascular and cost-effective benefits of pharmacist-management of cardiovascular-related diseases in populations who are under resourced. Evidence of sustained benefit from CVRR services would substantially enhance health outcomes and primary care service delivery.

The primary objective of this study was to assess the change in HbA1C from baseline after 3 to 5 months of follow up in the pharmacist-managed CVRR clinics. Secondary objectives included assessing the change in systolic blood pressure (SBP), diastolic blood pressure (DBP), low-density lipoprotein (LDL) cholesterol, and non-high-density lipoprotein (non-HDL) cholesterol at 3 to 5 months. Additional secondary objectives included demonstrating the

persistence of changes at 6–8 for the same clinical markers. Outcome measurements captured between 9 months and 36 months of follow-up were also described.

## Methods

This retrospective cohort chart review of the electronic medical record was deemed exempt by the Indiana University IRB, and informed written or verbal consent was not required. The review included eight pharmacist-managed CVRR clinics that are part of the county health system of a metropolitan area in Indiana, United States. This safety-net health system provides care to a primarily urban population of patients who are publicly insured, underserved, under-insured, and/or uninsured through a network of FQHCs. Pharmacists managing the CVRR clinics work under a collaborative practice agreement (CPA) for medication management of diabetes, hypertension, hyperlipidemia, and tobacco use. The first CVRR clinic was implemented in 2007, with seven CVRR clinics currently operating. An eighth CVRR clinic operated for two years before the pharmacist was moved to another clinic. A CPA is a legal document between a prescriber and a pharmacist, which grants additional privileges to the pharmacist [13]. The CPA is based on current clinical disease management guidelines (American Diabetes Association's Standards of Medical Care in Diabetes) and is reviewed and updated annually. A suggested treatment algorithm is included; however, the pharmacist may make clinical decisions based on the individual needs of the patient. The CPA allows CVRR pharmacists to initiate, change, or discontinue medications for all relevant conditions. Commonly used medication classes include: diabetes–metformin, sodium glucose cotransporter– 2 inhibitors, glucagon-like peptide-1 agonists, dipeptidyl peptidase 4 inhibitors, and insulin; hypertension–angiotensin converting enzyme inhibitors, diuretics, and calcium channel blockers; dyslipidemia–statins; and other medication classes to manage cardiovascular risk.

Patients are referred to the CVRR clinic if they are currently seen or have been seen at the FQHC in the past three years, an FQHC clinician refers them to CVRR services, or they have participated in group diabetes self-management education and support (DSMES) at the FQHC. The CVRR clinic does not require specific clinical criteria for eligible patients. Typically, CVRR patients will initially have an elevated HbA1C and may also have uncontrolled blood pressure and an elevated atherosclerotic cardiovascular disease (ASCVD) risk or lipid abnormalities. CVRR pharmacists usually meet with patients for a one-on-one, 30-minute appointment every four to six weeks. During this appointment, pharmacists review pertinent disease related lab values and monitoring parameters, provide education on disease management strategies, adjust cardiovascular pharmacologic therapies, and address barriers to disease state management optimization (i.e. prior authorization facilitation, referrals to other services, and other care coordination responsibilities). The frequency may be more or less often based on the needs of the patient. Throughout the time of working with the CVRR pharmacist, the patient will continue to meet with their primary care provider (PCP) for management of all medical conditions, as deemed necessary by the PCP. A patient will be discharged from the CVRR clinic once they maintain their individualized clinical goals (e.g.; HbA1C $\leq$ 7%) for three to six months, or if they no-show three scheduled appointments in a row despite attempts to contact or re-schedule them.

Data from eight CVRR clinics were included. The retrospective data were collected within the timeframe of January 1, 2015 through February 28, 2020. The process for CVRR data and outcome collection became standardized through the use of REDCap, a secure web-application database, across clinic sites on January 1, 2015. To allow for variability in time to first follow-up HbA1C after the initial CVRR visit, a window of three to five months was defined for primary outcome evaluation. Inclusion dates for each site were based on the date the clinical

pharmacy service was established and continued through six months prior to either the study end date (February 28, 2020) or six months prior to the clinical pharmacy service termination, to allow for sufficient time to appropriately capture data related to the primary outcome. Given the transition to telehealth-based care in late March 2020, REDCap measures were no longer consistently captured.

All patients seen by a CVRR pharmacist between January 1, 2015 and February 28, 2020 were evaluated for inclusion in the cohort. Patients were included in the study if they had two or more completed visits with the pharmacist at one of the eight clinic sites during the study period and the initial visit was on or before October 28, 2019, had a diagnosis of Type 2 diabetes mellitus (T2DM), and an initial HbA1C $\geq$ 8%. Patients were excluded if their initial HbA1C was < 8% or if they did not have a diagnosis of T2DM.

## Variables and statistical analysis

Data collected from the existing REDCap reports included CVRR visit information (clinic site, date of first CVRR visit, referring provider), basic patient demographics (age, gender, race, smoking status, initial statin therapy) and clinical markers (HbA1C, SBP, DBP, LDL, non-LDL, 10-year ASCVD risk). Because of the somewhat unpredictable nature of routine patient follow-up and clinical marker assessment, clinical marker data was captured according to how long after enrollment the clinical marker was assessed. The clinical marker data was assigned to three-month windows with data being allocated to whichever time period was closest to the end date of the clinical marker in question. For patients with two readings within the same window, clinical outcomes were averaged to reflect the level of control within that time period. In the event of a tie between two-time windows, data was allocated to the earlier time window.

The primary outcome of interest for this analysis was the change in HbA1C from baseline to the 3–5 month visit with the CVRR pharmacist. In order to determine if there was a clinically significant reduction of 0.5 points in the HbA1C between these paired time points with 80% power, allowable alpha error of 0.05, and a highly conservative standard deviation of 3.5, a sample size of 387 would be necessary [14]. With the more realistic expected standard deviation of 2, based on previous similar assessments within a population with more variability in their diabetes control, a sample size of 128 would be necessary to adequately test this hypothesis [15]. A minimum sample size of 400 was set to ensure that this investigation would have adequate power to identify significant differences in the primary outcomes.

The secondary outcomes included the change in SBP, DBP, and LDL between baseline and 3–5 months of follow-up. BP was classified by the highest SBP or DBP reading into the following categories: <120/80, 120-139/80-89, 140-159/90-99, and 160-179/100-109, and $\geq$180/110. In order to assess the persistence of these changes, for the patients with evaluable results the HbA1C, SBP, DBP, and LDL measurements at 6–8 months were also compared to the baseline results.

A paired t-test was utilized for all primary and secondary outcome comparisons with a *p<0.05* being deemed to be statistically significant. The mean of the differences between baseline clinical markers at 3–5 months and 6–8 months were also calculated and presented with the associated 95% confidence intervals. The mean difference was calculated by matching each baseline value to the same patient's subsequent outcome marker for each timepoint whenever follow-up data was available. Outcome measurements captured between 9 months and 36 months of follow-up were also visually illustrated, however, additional statistical analyses were not performed as the study was not adequately powered to assess those differences and the potential for selection bias after 9 months limits the utility of additional analyses. Patients remaining in the cohort for longer durations were also subject to selection bias as those who

were not lost to follow-up might be more likely to experience improvements in their care. Demographic characteristics were described using descriptive statistics and Stata 16® (College Station, TX) was utilized for all statistical analyses.

## Results

As seen in the study diagram in Fig 1, a total of 1,270 patients were assessed for eligibility with 631 ultimately being included in the analysis. A total of 445 patients were evaluated to test the primary outcome. Over the 36 months of evaluation, a total of 3,803 encounters were described in this analysis. At 36 months, 9.7% (n = 43) of the initial study population were eligible for evaluation.

As seen in Table 1, the mean (SD) age of participants was 54 (11) years. There was a slight predominance of female participants (56%) and African American patients were the main race (45%) within this cohort. Most patients had poorly controlled diabetes at baseline with a mean HbA1C of 10.9% [95%CI 10.7,11.0]. The other clinical markers of SBP, DBP, and LDL were mixed, with many patients achieving the desired targets for these markers prior to enrollment.

### Primary outcome

Compared to baseline, the HbA1C at three to five months was statistically significantly reduced to 9.3 (95%CI [9.1, 9.5]) with a mean difference of 1.6 percent ([95%CI -1.8, -1.4], $p<0.01$) after three to five months of CVRR care (Fig 2).

### Secondary outcomes

Reductions in HbA1C persisted at six to eight months (9.1 95%CI [8.9, 9.4], $p<0.01$) with a mean reduction of 1.8 percent ([95%CI -2.0, -1.5],). Patients who remained under the care of the CVRR pharmacist continued to contribute clinical marker data which demonstrated sustained reductions in HbA1C throughout the time period of evaluation (Fig 2).

There was a statistically significant drop from the mean baseline SBP of 131.7 (95%CI 130.3, 133.1) to a mean of 130.2 (95%CI 128.6–131.8, $p<0.05$) with a mean difference of -2.3 mmHg SBP at the three to five-month evaluation ([95%CI, -4.1, -0.4]). This statistically significant reduction was not noted at the 6 to 8 month evaluation of SBP (129.7, 95% CI 127.6–131.7, $p = 0.15$) (Fig 3). Inversely, the diastolic blood pressure at the three to five month time point (78.8, 95%CI [77.8, 79.7], $p = 0.05$, mean difference -1.2 95%CI [-2.3, -0.1]) was not statistically significantly reduced, while the six to eight month diastolic blood pressure was (77.7, 95%CI [76.4, 78.9], $p<0.05$, mean difference -2.1, 95%CI[-3.4,-0.7]). The analysis of LDL demonstrated statistically significant reductions at both the three to five month time point (-4.3 mg/dL, [95% CI -6.6, -1.1], $p<0.05$) and six to eight month time point (-7.75 mg/dL, [95%CI, -11.3,-4.2], $p<0.05$) (Fig 4). All clinical markers remained reduced or slightly elevated throughout the remaining period of evaluation for patients who continued to contribute results (Figs 2–4).

## Discussion

### Main findings of the study

The results of this study confirm that patients who consistently received care from pharmacist-managed CVRR services in FQHCs demonstrate sustained improvements in clinical outcomes related to cardiovascular risk for a minimum of six months. In the UKPDS study, patients who attained a lower HbA1C in the 10-year follow up had significant improvements in cardiovascular disease outcomes [3]. Patients who achieve and sustain glycemic control targets soon after diagnosis of diabetes are more likely to experience a reduction in cardiovascular risk [16].

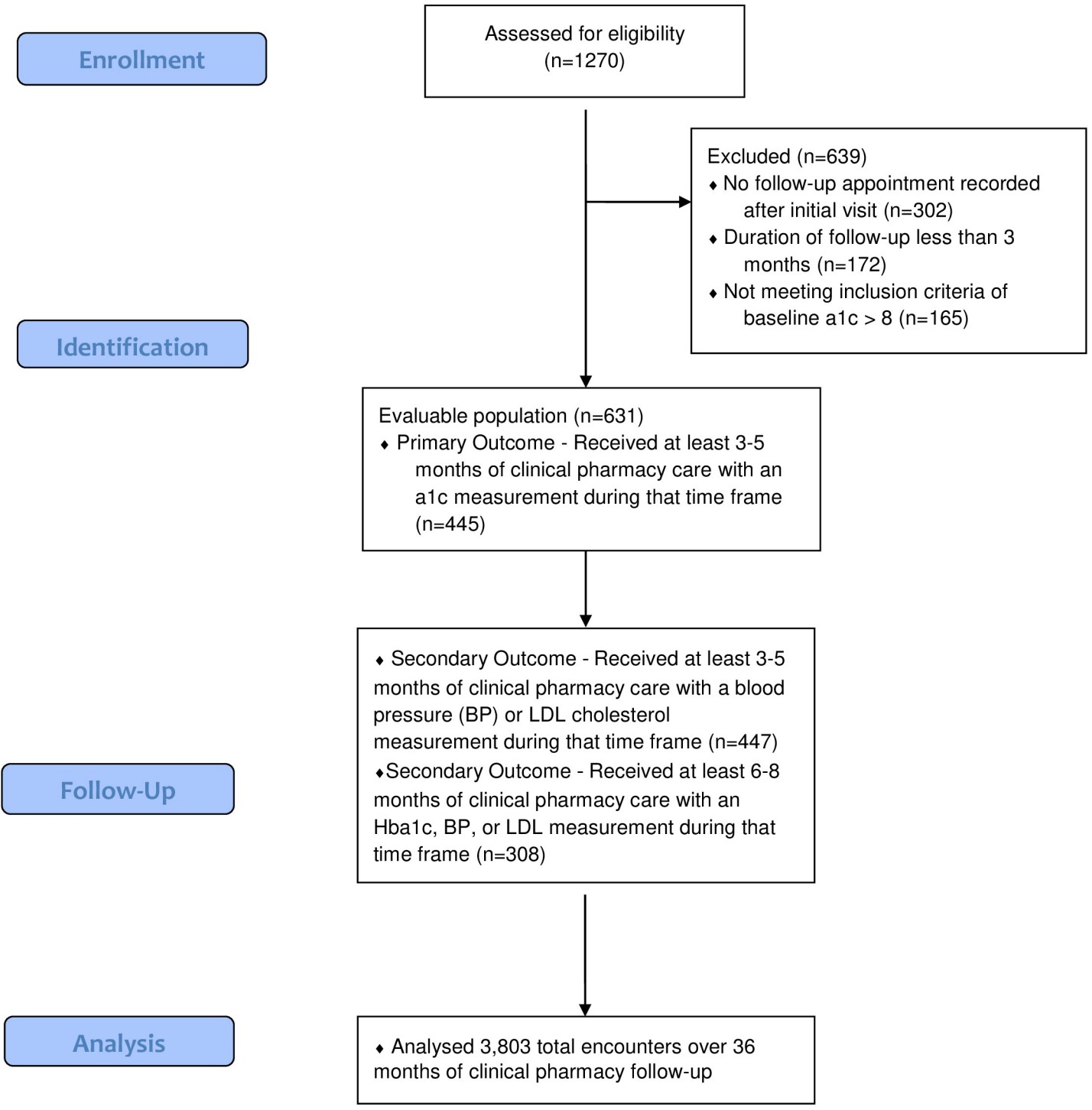

**Fig 1. Study diagram.** Patient characteristics.

The improvements in HbA1C demonstrated by the CVRR clinics herein exceed what would be expected from most pharmacologic diabetes treatments [17]. Despite significant improvements, the HbA1C of CVRR patients still remained above the expected goal of < 7% for most individuals in our study. However, given the reductions in HbA1C, LDL, and blood pressure,

**Table 1. Baseline demographic characteristics.**

| Characteristic | Results (n = 631) |
|---|---|
| Age, years, mean [SD] | 54.0 [11.0] |
| Age, years, n (%) | |
| 11–15 | 0 (0%) |
| 16–20 | 2 (< 1%) |
| 21–25 | 2 (< 1%) |
| 26–30 | 13 (2%) |
| 31–40 | 60 (10%) |
| 41–50 | 150 (24%) |
| 51–60 | 244 (39%) |
| 61–70 | 121 (19%) |
| 71–80 | 33 (5%) |
| >81 | 6 (< 1%) |
| Gender, n (%) | |
| Male | 275 (44%) |
| Female | 356 (56%) |
| Race/Ethnicity, n (%) | |
| Black or African American | 286 (45%) |
| White | 162 (26%) |
| Hispanic | 156 (25%) |
| Other | 27 (4%) |
| HbA1C, mean [SD] | 10.9% [1.9] |
| HbA1C, n (%) | |
| 8–10% | 268 (42%) |
| > 10% | 363 (58%) |
| Blood pressure, mmHg, mean [SD] | SBP 131.7 [18.1] / DBP 79.6 [10.8] |
| Blood pressure, mmHg, n (%) | |
| <120/80 | 129 (20%) |
| 120-139/80-89 | 299 (47%) |
| 140-159/90-99 | 145 (23%) |
| 160-179/100-109 | 48 (8%) |
| ≥180/110 | 10 (2%) |
| LDL Cholesterol, mg/dL, mean [SD] | 79.6 [10.8] |
| LDL Cholesterol, mg/dL, n (%) | |
| <70 mg/dL | 179 (28%) |
| 70–100 mg/dL | 185 (29%) |
| 100–129 mg/dL | 135 (21%) |
| ≥130 mg/dL | 132 (21%) |
| Clinic Location, n (%) | |
| Forest Manor | 47 (7%) |
| Grassy Creek | 31 (5%) |
| Midtown | 63 (10%) |
| North Arlington | 106 (17%) |
| Outpatient Care Center | 29 (5%) |
| Pecar | 52 (8%) |
| West 38th Street | 124 (20%) |
| West Side | 179 (28%) |

DBP: diastolic blood pressure; HbA1C: hemoglobin A1C; LDL: low-density lipoprotein; mg/dL: milligrams per deciliter; mmHg: millimeters of mercury; SBP: systolic blood pressure; SD: standard deviation.

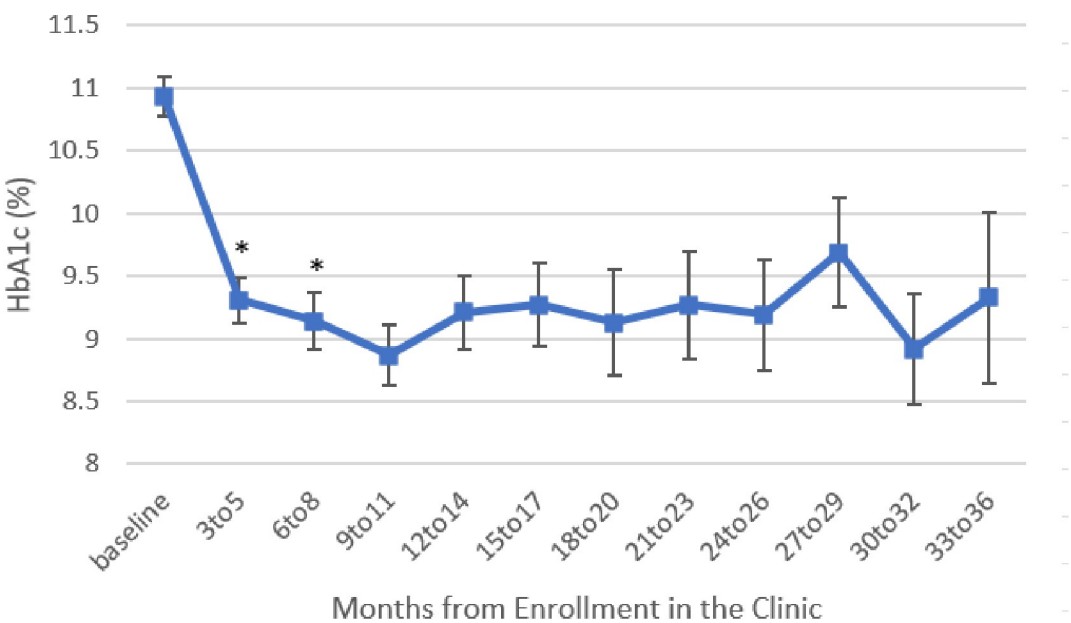

| Months | 0 | 3to5 | 6to8 | 9to11 | 12to14 | 15to17 | 18to20 | 21to23 | 24to26 | 27to29 | 30to32 | 33to36 |
|---|---|---|---|---|---|---|---|---|---|---|---|---|
| # of patients | 631 | 445 | 308 | 226 | 187 | 150 | 102 | 100 | 76 | 74 | 57 | 43 |
| Mean (95%CI) | 10.9 (10.8-11.1) | 9.3 (9.1-9.5)* | 9.1 (8.9-9.4)* | 8.9 (8.6-9.1) | 9.2 (8.9-9.5) | 9.3 (8.9-9.6) | 9.1 (8.7-9.6) | 9.3 (8.8-9.7) | 9.2 (8.7-9.6) | 9.7 (9.3-10.1) | 8.9 (8.5-9.4) | 9.3 (8.6-10.0) |
| Mean of Differences from baseline (95% CI) | | -1.6 (-1.8, -1.4) | -1.8 (-2.0, -1.5) | | | | | | | | | |

*Data from 9 through 36 months are shown as descriptive values without additional statistical evaluation.*

*= *p*<0.01 via paired t test comparison with baseline.
HbA1c: hemoglobin A1C

**Fig 2. Change in HbA1c from baseline to 36 months.**

our study demonstrates clinically significant decreases in cardiovascular risk in high risk populations traditionally associated with grave health disparities [18].

## What is already known on this topic

In 2017, the leading cause of death in Indiana, United States was heart disease, with rates high enough to be ranked 13th worst in the United States [19]. Patients who participate in the

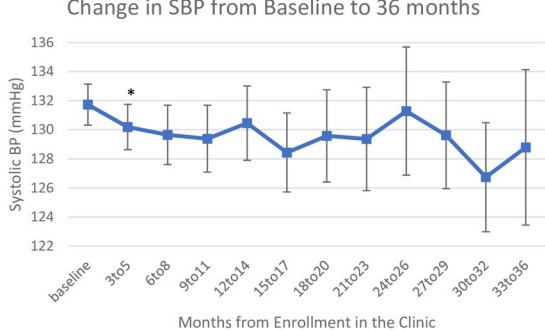

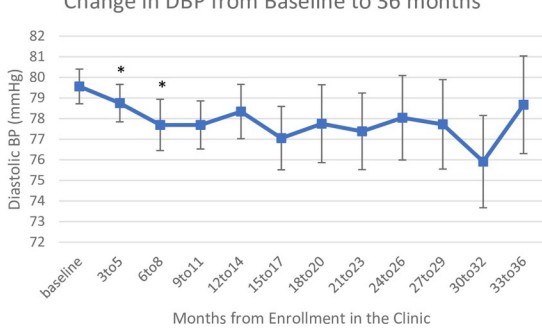

Fig 3. Change in systolic and diastolic blood pressure from baseline to 36 months.

| Months | 0 | 3to5 | 6to8 | 9to11 | 12to14 | 15to17 | 18to20 | 21to23 | 24to26 | 27to29 | 30to32 | 33to36 |
|---|---|---|---|---|---|---|---|---|---|---|---|---|
| # of patients con | 630 | 447 | 308 | 225 | 186 | 150 | 101 | 99 | 76 | 74 | 57 | 43 |
| Mean SBP (95%CI) | 131.7 (130.3–133.1) | 130.2 (128.6-131.8) | 129.7 (127.6-131.7) | 129.4 (127.1-131.7) | 130.5 (127.9-133.0) | 128.4 (125.7-131.2) | 129.6 (126.4-132.8) | 129.4 (125.8-132.9) | 131.3 (126.9-135.7) | 129.6 (125.9-133.3) | 126.7 (123.0-130.5) | 128.8 (123.4-134.1) |
| Mean of differences from baseline, SBP (95%CI) | | -2.3* (-4.1, -0.4) | -2.0 (-4.2, 0.3) | | | | | | | | | |
| Mean DBP (95% CI) | 79.6 (78.7-80.4) | 78.8 (77.8-79.7) | 77.7 (76.4-78.9) | 77.7 (76.5-78.9) | 78.3 (77.0-79.7) | 77.1 (75.5-78.6) | 77.8 (75.9-79.6) | 77.4 (75.5-79.2) | 78.0 (76.0-80.1) | 77.7 (75.5-79.9) | 75.9 (73.7-78.2) | 78.7 (76.3-81.0) |
| Mean of Differences from baseline, DBP (95%CI) | | -1.2* (-2.3, -0.1) | -2.1* (-3.4, -0.7) | | | | | | | | | |

*Data from 9 through 36 months are shown as descriptive values without additional statistical evaluation.*
*= *p*<0.05 via paired t test comparison with baseline.
DBP: diastolic blood pressure; mmHg: millimeters of mercury; SBP: systolic blood pressure

evaluated CVRR service represent some of the most vulnerable people in the country, in one of the states ranking worst in cardiovascular health. Populations served at FQHCs often face barriers due to social determinants of health, leading to disparities in care and poorer health

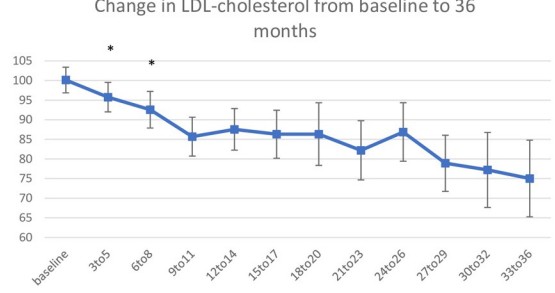

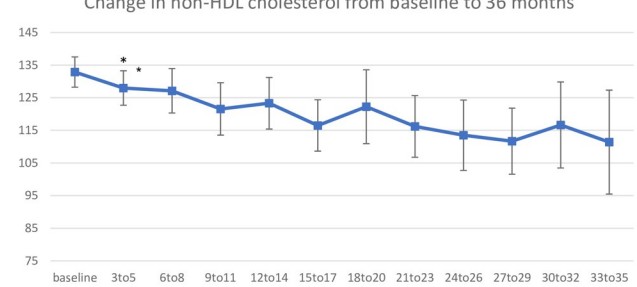

| Months | 0 | 3to5 | 6to8 | 9to11 | 12to14 | 15to17 | 18to20 | 21to23 | 24to26 | 27to29 | 30to32 | 33to36 |
|---|---|---|---|---|---|---|---|---|---|---|---|---|
| # of patients | 585 | 426 | 295 | 217 | 183 | 146 | 100 | 99 | 75 | 73 | 55 | 43 |
| Mean, 95%CI | 100.1 (96.9-103.4) | 95.8 (92.0-99.5) | 92.6 (87.9-97.2) | 85.7 (80.7-90.6) | 87.6 (82.3-92.8) | 86.3 (80.2-92.4) | 86.3 (78.4-94.3) | 82.2 (74.7-89.7) | 86.9 (79.4-94.3) | 78.9 (71.7-86.1) | 77.2 (67.6-86.8) | 75.0 (65.3-84.8) |
| Mean of differences from baseline, 95% CI | | -4.3* (-6.6,-1.1) | -7.6* (-11.3,-4.2) | | | | | | | | | |

Fig 4. Change in LDL from baseline to 36 months.

| Months | 0 | 3to5 | 6to8 | 9to11 | 12to14 | 15to17 | 18to20 | 21to23 | 24to26 | 27to29 | 30to32 | 33to36 |
|---|---|---|---|---|---|---|---|---|---|---|---|---|
| # of patients | 413 | 304 | 218 | 173 | 143 | 119 | 75 | 80 | 64 | 68 | 48 | 38 |
| Mean, 95% CI | 132.9 (128.2-137.5) | 128.0 (122.7-133.3) | 127.1 (120.3-133.9) | 121.6 (113.5-129.6) | 123.3 (115.4-131.2) | 116.5 (108.6-124.4) | 122.2 (110.9-133.6) | 116.2 (106.8-125.7) | 113.5 (102.7-124.3) | 111.7 (101.6-121.8) | 116.7 (103.5-129.8) | 111.4 (95.5-127.3) |
| Mean of differences from baseline, 95%CI | | -4.9 (-11.9, 2.2) | -5.7 (-14.0, 2.5) | | | | | | | | | |

*= *p*<0.05 via paired t test comparison with baseline. Statistical analysis was only performed to compare baseline
LDL: low-density lipoprotein; mg/dL: milligrams per deciliter; non-HDL: non-high-density lipoprotein

outcomes [18, 20]. Given the challenges of this setting, the results of our study are particularly noteworthy for demonstrating statistically and clinically significant improvements in cardiovascular risk. The number of participants exceeded the minimum number to meet power. Our data represent the quality of care provided by six pharmacists covering eight unique FQHC locations with over a decade of clinical service provision.

## What the study adds

Given the reduction in risk markers, such as HbA1C which have been observed in the CVRR clinic, there is strong potential for the reduction of major adverse cardiovascular events (MACE) for the high-risk patients seen in these clinics who maintain follow-up over time. Research consistently affirms the positive relationship between reducing HbA1C and lowering the risk of MACE, illuminating the potential for our work to significantly impact MACE outcomes [21]. Our data reflect one period of time across the patient's continuum of diabetes care. While HbA1C values during the study time period are above 7%, the overall trending decrease in HbA1C is clinically valuable. The UKPDS secondary analysis found that a 1% decrease in HbA1C is associated with a 35% reduction in microvascular outcomes, 18% reduction in myocardial infarction, and 17% reduction in all-cause mortality [3]. Additionally, the hope is that patients who are discharged from CVRR services due to achievement of clinical goals are able to sustain healthy outcomes due to the knowledge gained during their interactions with the CVRR pharmacist.

In 2017, diabetes expenditures in the United States were an estimated $327 billion, with a higher per person annual cost of $16,750 for those with diabetes compared to those without [22]. Each percentage reduction in HbA1C has been associated with a $685–950 per year savings in healthcare costs [4, 23]. While estimating cost savings was not an objective of this study, the observed reduction of HbA1C seen amongst patients receiving clinical pharmacist care in our clinics could result in considerable savings to the health system. Considering the vast financial impact of cardiovascular disease to society, it follows that the reduction in cardiovascular risk seen in this study is also commensurate with a significant reduction of cost. Given that the majority of patients enrolled in the CVRR service within our health-system are publicly insured, underserved, underinsured, and/or uninsured, any savings related to the care of these patients are also savings to the state and federal governments and taxpayers. Even accounting for the cost of providing the service, clinical pharmacists have demonstrated a positive return-on-investment [1].

In addition to direct revenue and cost avoidance, pharmacists also contribute to improvement in public health initiatives through their impact on quality measures. Goals for chronic disease management measures, especially those related to diabetes and hypertension, are routinely incentivized in value-based care programs by payors in order to assess the quality, safety, effectiveness, and efficiency of care provided [24]. Even in the FQHC setting, a large portion of patients covered by federally funded insurance programs, such as Medicare and Medicaid in the United States, are eligible to be included in value-based models. The improvements in clinical measures from pharmacist interventions within FQHCs can contribute to revenue through incentive payments and/or increases in per-member per-month reimbursement. Given the multiple accrediting agencies and variability in quality measures, healthcare organizations may prioritize different quality measures to evaluate performance. For example, in the United States, Healthy People 2030 and the Health Resources and Services Administration (HRSA) Uniform Data System Clinical Quality Measures specifically focus on a variety of objectives related directly to cardiovascular disease [25, 26]. Clinical pharmacists practicing in underserved areas must be aware of their specific community's prioritized quality measures

and goals. Our data highlight the need for comprehensive pharmacist integration within public health initiatives to optimize patient care and systemic efficiency.

Although not formally evaluated in this analysis, there are several potential explanations for success in diabetes management models involving pharmacists. In the United States, the majority of models using referral to a pharmacist service allow for additional one-on-one time dedicated to a focused set of related disease states, as opposed to managing both chronic and acute concerns in a shorter primary care visit [4]. Pharmacist CVRR visits are typically 30 minutes in length and are focused on CVRR-related conditions. In addition, the design of the pharmacist service typically allows for more frequent follow-up visits. At our institution the pharmacist schedules visits every four to six weeks, as compared to PCP visits which may occur every three to five months. More frequent follow-up offers opportunity to reduce therapeutic inertia [7]. Patients can follow with the CVRR service for a brief limited number of appointments or indefinitely based on individual characteristics and preferences, such as continued barriers impacting the ability to manage diabetes well or finding value in continued motivational touch points to maintain clinical goals. Also, given pharmacist training and expertise related to the medication distribution process, their skillset is optimal for ensuring medication access despite common barriers such as financial burden, insurance coverage changes, and manufacturer shortages or recalls. Due to the more frequent touchpoints and this background knowledge, the CVRR pharmacist team is able to prevent or promptly address access issues. Lastly, our pharmacist team possesses up-to-date, evidence-based expertise in clinical practice guidelines, pharmacotherapy algorithms, medication adverse effects, potential medication-related interactions, and other patient-specific considerations.

## Limitations of this study

Though the results of this study are robust, there are limitations. The time period used for the analysis did not account for temporary interruptions in the service. In instances of clinical pharmacist absence from the site, data may not have been collected. A number of patients were excluded from analysis due to only having one encounter with the clinical pharmacist. It is possible that there was benefit to these patients realized at the following PCP visit, however this data is not captured and was not evaluated. These patients likely have underlying differences from the patients who engaged with the clinical pharmacists, but due to the lack of detailed records, these differences are unevaluable. Possible reasons for this could include lack of engagement in their personal chronic disease state management, transportation challenges or other financial burden, and lack of perceived value in extra appointments above standard visits with their PCP. Selection bias is also a major potential limitation of this analysis as we could only analyze data from patients who had regular follow-up appointments. It is likely that these patients may have a stronger commitment to engage in healthier behaviors or perhaps an improved awareness about their disease process which may have contributed to the improvement in their clinical markers. It is also difficult to evaluate the independent impact of pharmacist managed services, because patients also maintain concurrent appointments with their primary care providers. In addition, clinical pharmacist care in these clinics is set up to allow patients to graduate out of care and return to routine care without continued follow-up by the clinical pharmacist. These transitions are not routinely documented within the electronic medical record used for this analysis making it difficult to differentiate which patients have been lost to follow-up versus those which appropriately do not require additional clinical pharmacist follow-up. Identifying individual reasons for lack of follow-up was beyond the capability of the electronic medical record report. While all included patients provided data from at least one follow-up visit, the variability in the enrollment date of patients considerably limited

the availability of follow-up data after 12 months as seen in Figs 2–4. While fasting and random blood glucose values would have added beneficial insight into overall diabetes management, including the incidence of hypoglycemia, these data were not included in this analysis due to the inconsistency in documentation and reporting of this information in the electronic medical record. HbA1C was identified as the best, standardized measure to report for the purposes of this study. However, underlying conditions that could potentially affect the accuracy of HbA1C values, such as anemia or thyroid conditions, were not collected. Furthermore, specific medication changes and dietary patterns were beyond the scope of this analysis and therefore not included. However, the pharmacists managing medications in this study follow evidence-based recommendations, prioritize first-line medications where appropriate, and limit barriers such as clinical inertia [7, 27]. Additional limitations of the study include retrospective design, lack of control group and evaluation within a single health care system.

## Conclusions

Our study augments the existing literature by demonstrating the health improvements associated with pharmacist-managed CVRR clinics within FQHCs in the United States. Our study identified the potential for service improvement to a network of FQHCs serving vulnerable populations, which warrants further evaluation. Additional studies exploring the augmentation or expansion of similar models may amplify the impact of pharmacist-managed CVRR services in primary care sites.

## Supporting information

**S1 Dataset.**
(XLSX)

## Author Contributions

**Conceptualization:** Jasmine D. Gonzalvo, Ashley H. Meredith, Andrew N. Schmelz, Lauren Pence, Jessica S. Triboletti.

**Data curation:** Sonak D. Pastakia, Madilyn Eberle, Lauren Pence.

**Formal analysis:** Sonak D. Pastakia, Michael Peters, Todd A. Walroth.

**Methodology:** Jasmine D. Gonzalvo, Ashley H. Meredith, Andrew N. Schmelz, Lauren Pence, Jessica S. Triboletti.

**Project administration:** Jasmine D. Gonzalvo, Ashley H. Meredith, Madilyn Eberle.

**Supervision:** Jasmine D. Gonzalvo, Ashley H. Meredith.

**Writing – original draft:** Jasmine D. Gonzalvo, Ashley H. Meredith, Sonak D. Pastakia, Michael Peters, Madilyn Eberle, Andrew N. Schmelz, Lauren Pence, Jessica S. Triboletti, Todd A. Walroth.

**Writing – review & editing:** Jasmine D. Gonzalvo, Madilyn Eberle, Andrew N. Schmelz, Todd A. Walroth.

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
