## [Decision Letter · Decision Letter 0]

21 Jun 2021

PONE-D-21-08257

Changes in Clinical Markers Observed from Pharmacist-Managed Cardiovascular Risk Reduction Clinics in Federally Qualified Health Centers: A Retrospective Cohort Study

PLOS ONE

Dear Dr. Gonzalvo,

Thank you for submitting your manuscript to PLOS ONE. After careful consideration, we feel that it has merit but does not fully meet PLOS ONE’s publication criteria as it currently stands. Therefore, we invite you to submit a revised version of the manuscript that addresses the points raised during the review process.

We look forward to receiving your revised manuscript.

Kind regards,

Vijayaprakash Suppiah, PhD

Academic Editor

PLOS ONE

Journal Requirements:

2. In ethics statement in the manuscript and in the online submission form, please provide additional information about the patient records/samples used in your retrospective study. Specifically, please ensure that you have discussed whether all data/samples were fully anonymized before you accessed them and/or whether the IRB or ethics committee waived the requirement for informed consent. If patients provided informed written consent to have data/samples from their medical records used in research, please include this information.

Additional Editor Comments (if provided):

Reviewers' comments:

Reviewer's Responses to Questions

**Comments to the Author**

1. Is the manuscript technically sound, and do the data support the conclusions?

Reviewer #1: Partly

Reviewer #2: Yes

Reviewer #3: Partly

2. Has the statistical analysis been performed appropriately and rigorously? 

Reviewer #1: No

Reviewer #2: Yes

Reviewer #3: Yes

3. Have the authors made all data underlying the findings in their manuscript fully available?

Reviewer #1: No

Reviewer #2: No

Reviewer #3: No

4. Is the manuscript presented in an intelligible fashion and written in standard English?

Reviewer #1: Yes

Reviewer #2: Yes

Reviewer #3: Yes

5. Review Comments to the Author

Reviewer #1: Manuscript PONE-D-21-08257 Thank you for giving us the opportunity to review the manuscript Title: “Changes in Clinical Markers Observed from Pharmacist-Managed Cardiovascular Risk Reduction Clinics in Federally Qualified Health Centers: A Retrospective Cohort Study” A manuscript in which the author described the efficacy of Reductions in hemoglobin A1c (HbA1C) and cardiovascular outcomes, the author recommended that health improvement of pharmacist-managed CVRR clinics. We have some points we would like to refer:

- In baseline characteristics the author needs to illustrate more data about other risk factors as existing cardiovascular disease, chronic kidney disease as the target control of HbA1c , systolic blood pressure ,BMI and LDL targets for each

- The mean follow up and p values should be enclosed in tables.

- Author should include the drugs used to control each risk factor.

- Non pharmacological methods for control risk factors as reduction of alcohol and salt consumption, body weight control and exercise recommendations.

Reviewer #2: The authors conducted a retrospective cohort study, describing the sustained mean reduction in HbA1C at 6 to 9 months of follow up through pharmacist managed CVRR clinic. This was also primary objective of the study. In general, the manuscript is well written, and provide optimism for pharmacist managed CVRR Clinic. However, the following noteworthy comments should be addressed to improve the impact of this manuscript.

1.The authors claim in the introduction section that a robust amount of evidence has established the impact of pharmacist managed CVRR clinics in reducing, and managing HbA1C, SBP/DBP and lipid levels. It is required that authors provide references for the previous studies which has provided these data for our readers. It will also be helpful if authors draw a meaningful comparison from these studies in the discussion section,

2.In the methods section, it will be helpful if authors could elaborate more on the responsibilities of pharmacists managing the CVRR clinic and services provided in these clinics.

3.In the result section, for increased clarity of the readers, the authors should also provide mean values of clinical risk markers (HbA1C, SBP, DBP, LDL, non-HDL) when significant at a given point of time during the follow up, in addition to providing the mean reduction, 95% CI and p value.

4.In the section “What Study Adds”, the authors should discuss the potential for reduction of major adverse cardiovascular events (MACE) with the sustained reduction in risk marker like HbA1C through the interventions of pharmacist managed CVRR clinic.

5.Lastly, in the limitation section, the authors should also acknowledge the selection bias since only patients who contributed to the follow up data by regularly following up with their appointments were included in the analysis. It is possible that these patients have healthy lifestyles, improved awareness about their disease process which might have also contributed to the reduction in their clinic markers (HbA1C).

Reviewer #3: The author demonstrates the benefit of pharmacist-managed cardiovascular risk reduction (CVRR) clinics mainly on HbA1c improvement by before-after study. This is the paper of important field, and the main result of primary outcomes is shown successfully as significant HbA1c improvement with sufficient statistical power as author has planned. However, because the present study is before-after study without control group, the results shown should be carefully assessed and interpreted. There also are other issues that should be amended.

<major comment="">

1. The author concluded that the pharmacist-managed CVRR clinic demonstrates sustained health improvements. However, because the statistical analysis is not made on data from 7 through 36 months according to Statistical Analysis in Methods section, the above-mentioned “sustained improvement” cannot be concluded or confirmed by the present study. Author should avoid and amend the conclusive description.

On the other hand, I agree with possible effects on sustained health improvement instinctively. The consideration of possible sustained health improvement itself is better to be left in Discussion section. Following viewpoints are addressed in Discussion section: (1) the data presented in this study are those of patients who continue to visit the pharmacist-managed CVRR clinic with poor follow-up rate of about 70% at 3 months and 7% at 36 months; (2) drop-out patients who is potentially poor outcome is better be addressed; and (3) the continued visits to pharmacist-managed CVRR clinic may possibly be just a predictor, not an effector.

2. Because this study is designed as before-after study, it is basically hard to confirm that the pharmacist-managed CVRR clinic cause HbA1c improvement at 3-6 months visit. It is desired to assess possible other contributing factors such as interventions by FQHC and/or primary care provider (PCP), especially at initiation.

3. Because the statistically evaluation is only made for data at baseline and 3-6 months visit, the continuous depiction of HbA1c and other clinical markers profile from baseline through 36 months may lead readers misinterpret. Therefore, author should clearly describe that the data from 7 through 36 months are just shown as descriptive values without statistical evaluation, or it might be desirable to create a Figure dedicated to the primary and secondary outcome evaluation at 3-6 months visit separately from Figure 2-4 if possible.

<minor comments="">

4. Because pharmacist-managed CVRR clinic is not a universal system and may have diversity among county even in United States, the system and the medical care that can be provided in Indiana county are summarized in Method section. In addition, the respective role of pharmacist-managed CVRR clinic and PCP in this system are recommended to be mentioned.

5. Drugs and treatment that can be handled in pharmacist-managed CVRR clinic should be clearly and concretely mentioned in Method section.

6. Since there seems to be a difference between the description in Statistical Analysis and the description in Result section and Figures, it is desirable to attach a protocol which is approved by the Ethics Committee and/or statistical analysis plan as Supplement.

7. Because reference 14 has no description of sample size calculation, author probably cite reference 14 to obtain the mean and standard deviation of HbA1c before and after treatment.　Author should demonstrate concrete HbA1c vales for sample size calculation. In addition, because “reduction of 0.5 points” would be “reduction of 0.5 percent”, and no description of page number of reference 14 in Reference section, author should correct them appropriately.

8. Because statistical analysis is not conducted to data at from 7 through 36 months visits, the description of analysis of data at 6-9 months visit should be avoided in Result section and Figures.

9. Author should clarify the handling of data at months in multiples of 3. For instance, if data at 6 months visit includes to 3-6 months period, description of 6-9 months should be corrected to 7-9 months. Similar correction should be made throughout the manuscript and Figures.

10. Author should demonstrate the rule of handling data in the following cases in Statistical Analysis in Methods section: (1) patient visits pharmacist-managed CVRR clinic more than twice during each 3 months period; and (2) data at 4 and at 5 months visits.

11. Author should amend followings in Figure 1: (1) because “Consort diagram” and “Allocation” are basically for terms of randomized clinical trial, it is better to be amended appropriately; and (2) because the fact that 631 patients are registered and 445 patients are able to evaluate at 3-6 months visit means that 186 patients dropped out from the study, the detail of drop out should be described.

12. The doubled figure of Non-HDL Cholesterol should be omitted in Figure 4.

13. Reference 7 is now searched usually. Correct description.

description.</minor></major>

6. PLOS authors have the option to publish the peer review history of their article (what does this mean?). If published, this will include your full peer review and any attached files.

Reviewer #1: No

Reviewer #2: No

Reviewer #3: No

---

## [Author Response · Author response to Decision Letter 0]

1 Dec 2021

Please see response to reviewers file for complete list of revisions.

---

## [Decision Letter · Decision Letter 1]

28 Mar 2022

PONE-D-21-08257R1Changes in Clinical Markers Observed from Pharmacist-Managed Cardiovascular Risk Reduction Clinics in Federally Qualified Health Centers: A Retrospective Cohort StudyPLOS ONE

Dear Dr. Gonzalvo,

Thank you for submitting your manuscript to PLOS ONE. After careful consideration, we feel that it has merit but does not fully meet PLOS ONE’s publication criteria as it currently stands. Therefore, we invite you to submit a revised version of the manuscript that addresses the points raised during the review process.

We look forward to receiving your revised manuscript.

Kind regards,

Vijayaprakash Suppiah, PhD

Academic Editor

PLOS ONE

Journal Requirements:

Reviewers' comments:

Reviewer's Responses to Questions

**Comments to the Author**

1. If the authors have adequately addressed your comments raised in a previous round of review and you feel that this manuscript is now acceptable for publication, you may indicate that here to bypass the “Comments to the Author” section, enter your conflict of interest statement in the “Confidential to Editor” section, and submit your "Accept" recommendation.

Reviewer #1: All comments have been addressed

Reviewer #2: All comments have been addressed

Reviewer #3: (No Response)

2. Is the manuscript technically sound, and do the data support the conclusions?

Reviewer #1: Partly

Reviewer #2: Yes

Reviewer #3: Yes

3. Has the statistical analysis been performed appropriately and rigorously? 

Reviewer #1: No

Reviewer #2: (No Response)

Reviewer #3: Yes

4. Have the authors made all data underlying the findings in their manuscript fully available?

Reviewer #1: No

Reviewer #2: (No Response)

Reviewer #3: No

5. Is the manuscript presented in an intelligible fashion and written in standard English?

Reviewer #1: No

Reviewer #2: (No Response)

Reviewer #3: Yes

6. Review Comments to the Author

Reviewer #1: -Risk factors as presence of cardiovascular disease, cerebrovascular disease or peripheral vascular disease should be included in the baseline characteristics

- fasting and random blood sauger measurement should be mentioned as a standard of blood sugar control

-presence of hypoglycemia not mentioned.

-the types of medication insulin, oral hypoglycemic or dietary control are not clear.

-many underlying conditioned as anemia and hyperthyroidism could affect the accuracy of HBA1C thus, it should be excluded if present.

Reviewer #2: (No Response)

Reviewer #3: The author demonstrates the benefit of pharmacist-managed cardiovascular risk reduction (CVRR) clinics. This is the paper of important field.

<comments>

1. The author may mention the possibility that the patient insurance states might affect the results especially in underinsured patients. Since uninsured patients may tend to avoid expensive medical care, it might be pronounced the importance of being able to improve and maintain 2% reduction in HbA1c even the value itself is considered as poor HbA1c control situation.

2. The follow-up rate at 36 months should be mentioned in Result section.

3. Regarding Table 1, the author should clearly state more detailed rule of classifying blood pressure in the case that the systolic and diastolic blood pressure belong to different subgroups such as 130/70 mmHg. The author should state the rule to treat blood pressure in Statistical Analysis section.

4. As the caption of Fig.1, “Consort Diagram. Patient characteristics” is inappropriate. The author should consider to change to “Study diagram”

5. “Secondary outcomes” in line 194 should be amended as subheading using Bold font and appropriate letter size.

6. “A1C” in line 243 should be corrected to “HbA1C”.

7. According to the response to reviewers and revised manuscript, the statistical analysis is not conducted to data at from 9 through 36 months visits. The description of footnote in Figure 2 and 3 should be corrected so that it does not contradict.

8. According to the description that the additional statistical analyses were not performed for the outcome measurements captured between 9 months and 36 months, the time window of secondary endpoint should be 6-8 months and the time window of primary endpoint should be 3-5 months. The author should check again and correct description if necessary.

9. The author stated “The external validity of this study is strengthened by the multi-site nature of the data” in line 236. However, because no external validation was conducted in this study, this sentence should be omitted.</comments>

7. PLOS authors have the option to publish the peer review history of their article (what does this mean?). If published, this will include your full peer review and any attached files.

Reviewer #1: No

Reviewer #2: No

Reviewer #3: No

---

## [Author Response · Author response to Decision Letter 1]

7 Apr 2022

The response to reviewers file contains all of the relevant information.

---

## [Decision Letter · Decision Letter 2]

6 May 2022

PONE-D-21-08257R2

Changes in Clinical Markers Observed from Pharmacist-Managed Cardiovascular Risk Reduction Clinics in Federally Qualified Health Centers: A Retrospective Cohort Study

PLOS ONE

Dear Dr. Gonzalvo,

Thank you for submitting your manuscript to PLOS ONE. After careful consideration, we have decided that your manuscript does not meet our criteria for publication and must therefore be rejected.

I am sorry that we cannot be more positive on this occasion, but hope that you appreciate the reasons for this decision.

Kind regards,

Vijayaprakash Suppiah, PhD

Academic Editor

PLOS ONE

Reviewers' comments:

Reviewer's Responses to Questions

**Comments to the Author**

1. If the authors have adequately addressed your comments raised in a previous round of review and you feel that this manuscript is now acceptable for publication, you may indicate that here to bypass the “Comments to the Author” section, enter your conflict of interest statement in the “Confidential to Editor” section, and submit your "Accept" recommendation.

Reviewer #1: (No Response)

Reviewer #2: All comments have been addressed

Reviewer #3: All comments have been addressed

2. Is the manuscript technically sound, and do the data support the conclusions?

Reviewer #1: Partly

Reviewer #2: (No Response)

Reviewer #3: Partly

3. Has the statistical analysis been performed appropriately and rigorously? 

Reviewer #1: No

Reviewer #2: (No Response)

Reviewer #3: No

4. Have the authors made all data underlying the findings in their manuscript fully available?

Reviewer #1: No

Reviewer #2: (No Response)

Reviewer #3: No

5. Is the manuscript presented in an intelligible fashion and written in standard English?

Reviewer #1: Yes

Reviewer #2: (No Response)

Reviewer #3: Yes

6. Review Comments to the Author

Reviewer #1: i reject the manuscript "changes in Clinical Markers Observed from Pharmacist-Managed Cardiovascular Risk Reduction Clinics in Federally Qualified Health Centers: A Retrospective Cohort Study " as no data support the results moreover, all authors comments are not covered appropriately despite multiple revisions.

Reviewer #2: No further comments from my end. Thank you..........................................................

Reviewer #3: The author demonstrates the benefit of pharmacist-managed cardiovascular risk reduction (CVRR) clinics. The collected data itself is surely valuable. However, the several descriptions are still incoherent and irrational which need amendment. In addition, the several statements regarding peculiar study population and unstrict diabetes mellitus (DM) management in Limitation section should be mentioned and addressed in Discussion section. Otherwise, the reader may be leaded misinterpretation.

<comments>

1. The author’s response to last reviewer comment 1 for manuscript PONE-D-21-08257.R1 is seemed to be insufficient. The followings should be amended or addressed appropriately: (1) “the majority of patients enrolled in the CVRR service within our health-system are uninsured or have public insurance” (line 255-256, page 13) should be amended to “the patients enrolled in the CVRR service within our health-system are underserved, underinsured and/or uninsured” as stated in Methods section (line 86, page 4); and (2) the poor HbA1c value of around 9.0 to 9.5 % without further improvement throughout follow-up period in study population should be mentioned and addressed in Discussion section. I recommend to address this issue by citing the paper regarding diabetes mellitus (DM) in uninsured patients (for instance, Talley et al. Popul Health Manag. 2018;21(5):373, and so on), and suggest to add the opinion that “Even if it is judged as poor control according to the DM guidelines, even one percent improvement in HbA1c would probably be valuable for uninsured diabetics who may have not visited medical institution”.

2. The small follow-up rate at CVRR clinic is another issue to be clarified and addressed as mentioned in reviewer comment for PONE-D-21-08257. The followings should be addressed and mentioned: (1) According to the description in Methods section (line 115-116), patients who maintains their clinical goals for 3-6 months will be discharged from the CVRR clinic. The ratio or cumulative number of patients who achieve and maintain their clinical goals for 3-6 months and are discharged from CVRR clinic should be shown. This information will be expected to reinforce the benefit of CVRR clinic.; (2) The ratio or cumulative number of drop-out patients who no-show three scheduled appointments in a row is desired to be shown.; and (3) To avoid misunderstanding, the peculiar characteristics of patients who continue to visit CVRR clinic longer than 9 months without discharge or drop-out should be mentioned again in Discussion section. In addition, if the limited health insurance of this peculiar population may be the cause of relatively poor HbA1c control by avoidance of sufficient but expensive medical treatment, the author may mention this possibility in Discussion section.

3. The DM guideline which is usen in CVRR clinic should be specified. Similarly, the clinical goal to discharge from CVRR clinic should be stated.

4. Because the sample size calculated in this study is only for primary outcome according to the description in Statistical analysis, the description “A minimum sample size of 400 was ultimately selected to ensure that this investigation would have adequate power to identify clinically significant differences in both the primary and secondary outcomes” (page 7,line 153-155) should be amended to “A minimum sample size of 400 was set to ensure that this investigation would have adequate power to identify significant differences in the primary outcomes”.

5. Because this study is not designed to determine if the interventions provided by CVRR pharmacists are responsible for the HbA1c improvements which have been observed in CVRR clinic, the description “such as HbA1C, through the intervention of the CVRR pharmacist,” (page 12, line 242-243) should be amended to “such as HbA1C which have been observed in CVRR clinic.

6. Because this study is not demonstrated any data regarding medical costs, the description “Our study identified the potential for service improvement through significant cost savings to a network of FQHCs serving vulnerable population” (page 16, line 332-334, in Conclusions section) should be amended to “Our study identified the potential for service improvement to a network of FQHCs serving vulnerable population”. The future study to evaluate cost savings in CVRR clinic might be suggested in Discussion section if the author would like to mention.</comments>

7. PLOS authors have the option to publish the peer review history of their article (what does this mean?). If published, this will include your full peer review and any attached files.

Reviewer #1: No

Reviewer #2: No

Reviewer #3: No

- - - - -

---

## [Author Response · Author response to Decision Letter 2]

20 Jul 2022

Please see attached 'Response to Reviewers' document for comprehensive responses addressing all feedback.

---

## [Decision Letter · Decision Letter 3]

15 Feb 2023

PONE-D-21-08257R3

  Changes in Clinical Markers Observed from Pharmacist-Managed Cardiovascular Risk Reduction Clinics in Federally Qualified Health Centers: A Retrospective Cohort Study PLOS ONE

Dear Dr. Gonzalvo,

First of all, I would like to apologize for the long time the article has been under review. I just accepted this article as an Academic Editor and hope the reviewing process and final decision is achieved soon.

Your research is of great interest because it highlights how healthcare professionals (in this case the pharmacist) can impact positively on health outcomes of patients with diabetes mellitus.

You have addressed commentaries by reviewers and the article has clearly improved. Comments from last reviewers are mostly favourable.

After careful consideration, we feel that it has merit but does not fully meet PLOS ONE’s publication criteria as it currently stands. My decision is Minor review, which means that with some minor changes the article could be accepted. Therefore, we invite you to submit a revised version of the manuscript that addresses the points raised during the review process

Some comments:

- Introduction: It would be of interest to reinforce the need of more evidence that justifies the study. The sentence “A robust amount of evidence has established the impact of pharmacists in reducing and managing HbA1C levels, systolic and diastolic blood pressure, and lipid levels to mitigate cardiovascular risks [10-12]” is wrongly pointing in the opposite direction as you say there is a lot of evidence on several topics, including diabetes (which would not justify your study), which is not totally correct and is not relating to the references used (that show only some evidence on DM). This should be rewritten to express the idea that there is some evidence of interest showing that pharmacists’ interventions may result in improvements of several cardiovascular variables but further research is needed specifically in the field of DM management, and that it would be of interest or specially needed for specific population subgroups at risk (e.g. the population in your study). As for the next phrase “However…” please remove the word “however”.

- As for the “Statistical analysis” subsection, it starts enumerating and defining the variables collected and related issues (which is not statistical analysis). Thus, it would improve if this title is changed for “Variables and statistical analysis”

- Also, in the “statistical analysis” subsection, there is a minor error that should be changed: “140-159/90-99, and 160-179/100-110, and >180/110.”  “140-159/90-99, 160-179/100-110, and >180/110.”

- Results section. Table 1… in the first cell of the second column you write “n (%)” but below you also show variables with mean and (SD). Thus, you should change it for something generic such as “Total sample (n=your sample size)” and for each variable include (n, %). E.g.: "Gender (n, %)". Also, for the variables where you indicate (mean, SD) and (n,%), you should name the variable in a manner that the reader can easily identify what your are showing… E.g.: one row with “Age (mean, SD)” and another row with “Age (n, %). Please also change “characteristic” to a more appropriate wording.

As for “reviewer 1” comments (see the complete commentaries in the reviewer response) I would recommend you to address them as they would improve the article with little effort. In summary:

1. Methods section of the abstract

Please change “All patients seen by a CVRR pharmacist over a 5-year period were evaluated” to something similar to the information you provide in the article… e.g.: “Data were collected from patients seen by a CVRR pharmacist within the timeframe of January 1, 2015 through February 28, 2020” (or something similar). ALSO, add something regarding “timeline” of the analysis e.g. “HbA1C from baseline after 3 to 5 months was assessed with pared t-tests analysis. Other clinical variables were assessed and additional analysis were performed at 6-8 months. Additional results are reported between 9 months and 36 months of follow up”. Something in this line would help the reader to understand what you did.

2. Dipeptidyl peptidase 4 inhibitors

Please state in the introduction section that dipeptidyl peptidase 4 inhibitors were used (if they were) or alternatively mention this as a limitation in the discussion as the restriction of therapy available to patients may impact on the results compared with other practice.

3. In Discussion section, Main findings of the study paragraph

Please mention the finding the reviewer commented.

4. Same, follow reviewer advice.

5. The reviewer states the importance of follow-up problems. Please state a sentence in the results sections of the ABSTRACT with this regard. E.g. The follow-up losses were XX% at 3-5 months and YY at ZZ months. Also, in the conclusion section of the abstract you should state something such as “the great proportion of loss to follow-up is a limitation of this study to be considered”.

Also, as recommended by one of the reviewers, please explain these problems and their implications in the discussion section as a limitation.

We look forward to receiving your revised manuscript.

Kind regards,

Noe Garin, Ph.D.

Academic Editor

PLOS ONE

Journal Requirements:

1. Please upload a Response to Reviewers letter which should include a 
point by point response to each of the points made by the Editor and / or Reviewers. (This should be uploaded as a 'Response to Reviewers' file type.) Please follow this link for more information: http://blogs.PLOS.org/everyone/2011/05/10/how-to-submit-your-revised-manuscript/

Additional Editor Comments (if provided):

<comments>

Reviewers' comments:

Reviewer's Responses to Questions

 </comments>

<comments>

**Comments to the Author**

1. If the authors have adequately addressed your comments raised in a previous round of review and you feel that this manuscript is now acceptable for publication, you may indicate that here to bypass the “Comments to the Author” section, enter your conflict of interest statement in the “Confidential to Editor” section, and submit your "Accept" recommendation.

Reviewer #1: All comments have been addressed

Reviewer #3: (No Response)

Reviewer #4: All comments have been addressed

2. Is the manuscript technically sound, and do the data support the conclusions?

Reviewer #1: Yes

Reviewer #3: Partly

Reviewer #4: Yes

3. Has the statistical analysis been performed appropriately and rigorously? 

Reviewer #1: N/A

Reviewer #3: Yes

Reviewer #4: Yes

4. Have the authors made all data underlying the findings in their manuscript fully available?

Reviewer #1: Yes

Reviewer #3: Yes

Reviewer #4: No

5. Is the manuscript presented in an intelligible fashion and written in standard English?

Reviewer #1: No

Reviewer #3: Yes

Reviewer #4: Yes

6. Review Comments to the Author

Reviewer #1: i accept the manuscript PONE-D-21-08257R3 titled " Changes in Clinical Markers Observed from Pharmacist-Managed Cardiovascular Risk Reduction Clinics in Federally Qualified Health Centers: A Retrospective Cohort Study"

for publication.

Reviewer #3: The author intends to demonstrate the benefit of pharmacist-managed cardiovascular risk reduction (CVRR) clinics. Although the collected data itself is surely valuable, the adopted population is unique because of non-negligible poor follow-up rate (70% at 3-5 months; 50% at 6-9 months; and 30% at 12-15 months). The author should mention that the findings in the present study was the observation in such relatively minor population who continuously visit CVRR clinic, otherwise the reader can be misinterpreted.

<comments>

1. In Abstract, the author described “All patients seen by a CVRR pharmacist over a 5-year period were evaluated” without any description regarding statistically analysis, while the author described only regarding the findings at 3-5 months and at 6-8 months. This inconsistent and insufficient description should be amended.

2. In Introduction section: Are dipeptidyl peptidase 4 inhibitors not allowed in CVRR clinic?

3. In Discussion section, Main findings of the study paragraph: Regarding the description “The results … six months”, the author should mention that this finding was observed in about half of the populations who continuously attended CVRR clinic.

4. In Discussion section, What the study add paragraph: Regarding the description “Given the reduction … these clinics.”, the author should also mention that this potential was for about half of the populations who continuously attended CVRR clinic.

5. As the reviewer mentioned continuously, the small follow-up rate at CVRR clinic is non-negligible issue. The followings should be addressed and mentioned: (1) the author’s opinion regarding the benefit of education in CVRR clinic for the discharged patients who achieved their clinical goals; and (2) assessment of poor follow-up rate in Discussion section (the reviewer recommends to move relevant issue from Limitation section)</comments>

Reviewer #4: Well done on this important research study with insightful findings. You have clearly acknowledged all the limitations with your paper which should prompt further research.

7. PLOS authors have the option to publish the peer review history of their article (what does this mean?). If published, this will include your full peer review and any attached files.

Reviewer #1: No

Reviewer #3: No

Reviewer #4: **Yes: **

While revising your submission, please upload your figure files to the Preflight Analysis and Conversion Engine (PACE) digital diagnostic tool, https://pacev2.apexcovantage.com/. PACE helps ensure that figures meet PLOS requirements. To use PACE, you must first register as a user. Registration is free. Then, login and navigate to the UPLOAD tab, where you will find detailed instructions on how to use the tool. If you encounter any issues or have any questions when using PACE, please email PLOS at figures@plos.org. Please note that Supporting Information files do not need this step.</comments>

---

## [Author Response · Author response to Decision Letter 3]

18 Feb 2023

Please see attached Response to Reviewers Table

---

## [Editor Report · Decision Letter 4]

28 Feb 2023

Changes in Clinical Markers Observed from Pharmacist-Managed Cardiovascular Risk Reduction Clinics in Federally Qualified Health Centers: A Retrospective Cohort Study

PONE-D-21-08257R4

Dear Dr. Gonzalvo,

We’re pleased to inform you that your manuscript has been judged scientifically suitable for publication and will be formally accepted for publication once it meets all outstanding technical requirements.

Kind regards,

Noe Garin, Ph.D.

Academic Editor

PLOS ONE

---

## [Editor Report · Acceptance letter]

3 Mar 2023

PONE-D-21-08257R4 

Changes in clinical markers observed from pharmacist-managed cardiovascular risk reduction clinics in federally qualified health centers: a retrospective cohort study 

Dear Dr. Gonzalvo:

I'm pleased to inform you that your manuscript has been deemed suitable for publication in PLOS ONE. Congratulations! Your manuscript is now with our production department. 

Kind regards, 

on behalf of

Dr. Noe Garin 

Academic Editor

PLOS ONE